# Non-Isothermal Crystallization Kinetics of Poly(ethylene glycol) and Poly(ethylene glycol)-B-Poly(ε-caprolactone) by Flash DSC Analysis

**DOI:** 10.3390/polym13213713

**Published:** 2021-10-27

**Authors:** Xiaodong Li, Meishuai Zou, Lisha Lei, Longhao Xi

**Affiliations:** Beijing Institute of Technology, School of Materials Science and Engineering, Beijing 100081, China; 3220201147@bit.edu.cn (L.L.); 1120180368@bit.edu.cn (L.X.)

**Keywords:** PEG-PCL, non-isothermal crystallization, flash differential scanning calorimeter

## Abstract

The non-isothermal crystallization behaviors of poly (ethylene glycol) (PEG) and poly (ethylene glycol)-b-poly(ε-caprolactone) (PEG-PCL) were investigated through a commercially available chip-calorimeter Flash DSC2+. The non-isothermal crystallization data under different cooling rates were analyzed by the Ozawa model, modified Avrami model, and Mo model. The results of the non-isothermal crystallization showed that the PCL block crystallized first, followed by the crystallization of the PEG block when the cooling rate was 50–200 K/s. However, only the PEG block can crystallize when the cooling rate is 300–600 K/s. The crystallization of PEG-PCL is completely inhibited when the cooling rate is 1000 K/s. The modified Avrami and Ozawa models were found to describe the non-isothermal crystallization processes well. The growth methods of PEG and PEG-PCL are both three-dimensional spherulitic growth. The Mo model shows that the crystallization rate of PEG is greater than that of PEG-PCL.

## 1. Introduction

Differential scanning calorimetry (DSC) is commonly used to study the crystallization behavior of polymers [1,2,3,4,5]. However, the actual processing technology of polymer materials, including injection molding, blown film, and spinning, the actual cooling rate is far greater than the maximum cooling rate that can be achieved by conventional DSC. Therefore, it is difficult to use DSC to simulate the crystallization environment of polymers in the actual processing. In order to solve the above problems, people have turned to the development of a new type of calorimeter with miniaturization, high sensitivity, and high temperature resolution, and with ultra-fast temperature rise and fall rates. In 2010, METTLER TOLEDO used the ceramic substrate chip sensor MultiSTAR USF1 (XI-400) to manufacture the first commercial power-compensated fast scanning chip calorimeter Flash DSC 1 [6,7,8]. Fast-scan chip-calorimetry (FDSC) has been developed in recent years. At present, the heating rate of flash DSC2+ can reach 3,000,000 K/s, and the cooling rate can reach 2,400,000 K/s. The ultra-fast heating and cooling rate of FDSC is mainly due to the chip sensor used in the instrument [9]. The chip sensor consists of two independent calorimeters, corresponding to the sample and reference sample areas. The calorimeter has a double-layer film structure, which is a silicon nitride film and a silicon oxide film dielectric layer. The total thickness of the double-layer film is about 2 p.m. The center of the calorimeter film is the temperature control area with a diameter of about 0.5 mm. With the help of a microscope, the experimenter can place a sample of tens of microns into the central area of the calorimeter to obtain uniform temperature control.

The biggest advantage of FDSC lies in its ultra-high cooling rate, which is controllable in the preparation of specific structures and specific thermal history materials. In addition, its ultra-fast heating rate can greatly inhibit the structural rearrangement during the heating process. This phenomenon shows its unique advantages in the crystallization research of many polymer materials. For example, Cavallo uses FDSC to compare the isothermal crystallization kinetics of isotactic polypropylene, propylene/butene copolymer, and propylene/hexene copoly-mer in the range of 0–90 °C [10]. Chen et al. studied the change in the crystal morphology of polyvinylidene fluoride-chlorotrifluoroethylene [P(VDF-CTFE)] by adding a nu-cleating agent via FDSC [11].

Currently, efforts are focused on understanding the non-isothermal crystallization behavior of polymers, because polymer processing is usually carried out under non-isothermal conditions. Different methods have been developed to evaluate non-isothermal crystallization based on DSC experimental data, such as Ozawa model, Avrami model, Mo model, etc., [12,13,14]. Some of these methods can be better explained in the DSC experimental data obtained at a lower cooling rate [15,16,17,18]. However, it is a mystery whether it can be better explained when it is applied to the FDSC experimental data of the ultra-fast cooling rate.

Poly(ethylene glycol)-b-poly(ε-caprolactone) (PEG-PCL) has good biocompatibility, biodegradability, and is easy to synthesize. These properties make it have great potential in drug delivery systems [19,20,21,22,23]. Many properties of this type of block copolymers, such as drug permeability, degradation performance and mechanical properties, are significantly affected by their crystalline behavior and aggregate structure [24,25,26,27,28,29]. Therefore, the study of their crystallization behaviors has important theoretical and practical significance. Bogdanov et al. characterized the thermal properties of three PCL-b-PEG copolymers [30]. It was concluded that the PCL blocks crystallize first, which determines the total copolymer structure and leads to imperfect crystallization of the PEG blocks. Shiomi et al. observed the morphology of spherulites of PCL-PEG-PCL triblock copolymers with different block lengths [31].

In this report, we selected the homopolymer PEG and block copolymer PEG-PCL to study their non-isothermal crystallization behavior under a high cooling rate measured by FDSC. The Ozawa model, Avrami model, and Mo model are used to evaluate the non-isothermal crystallization behavior of PEG and PEG-PCL at high cooling rates, and compare the results with the non-isothermal crystallization results by DSC.

## 2. Materials and Methods

### 2.1. Materials

The PEG (Mw = 5000), PCL (Mw = 5000), and PEG-PCL (50%/50%mol, Mw = 10,000) used in this experiment were purchased from Guangzhou TanSh technology Co., Ltd (Guangzhou, China).

### 2.2. Test Instrument

The flash differential scanning calorimeter used was Flash DSC2+ (manufactured by Mettler-Toledo Company). At the beginning of all experiments, each sensor was adjusted and calibrated. All the detections were conducted under a nitrogen atmosphere with a constant flow rate of 80 mL/min.

The bulk samples were cut into small pieces (about 10–100 ng) and then were transferred to the sensor center within an area of (0.2–0.5) mm^2^ under the microscope. In order to maintain good contact between the sample and the sensor, a pre-melting procedure was adopted, with repeated heating and cooling procedures; the heating and cooling rates of pre-melting were both 10 K/s.

### 2.3. Non-Isothermal Crystallization Process

The non-isothermal crystallization was performed as follows: the samples were heated up to 120 °C higher than the melting point of the samples at a heating rate of 2000 K/s, this temperature was maintained for 1 s to obtain the equilibrium of the melt. Then the samples were cooled to −80 °C at different cooling rates, which is lower than its glass transition temperature. The temperature program is shown in Figure 1.

## 3. Results and Discussion

### 3.1. Crystallization of Samples under Various Cooling Rates

Figure 2 plots the cooling curves of PEG and PEG-PCL under various cooling rates measured by FDSC. In Figure 2a, an obvious crystallization peak can be observed when the cooling rate is low. The crystallization exothermic peak of PEG becomes broad and weak when the cooling rate reaches 800 K/s. PEG is completely inhibited from crystallization when the cooling rate exceeds 1000 K/s. In Figure 2b, double crystallization peaks appear on the cooling curve of PEG-PCL when the cooling rate is 50 K/s, and 100 K/s, which shows that both the PEG block and the PCL block can crystallize when the cooling rate is low. This is similar to the results measured by Hiroki et al. using DSC; they found that PCL crystallized first, followed by the crystallization of PEG with preservation of the PCL crystal lamellar structure [29]. However, the double crystallization peak becomes a single peak when the cooling rate reaches 200 K/s. The crystallization peak of PEG-PCL disappears when the cooling rate continues to increase to 1000 K/s.

The relative crystallinity *X*(*T*) can be obtained from the ratio of the area of the exothermic peak of the crystallization at the temperature of the curve to the area of the entire crystallization peak when the crystallization is completed. The formula is as follows [12]:(1)$$X(T)=\frac{\int_{T_{0}}^{T}(dH_{c}/dT)dT}{\int_{T_{0}}^{T_{\infty }}(dH_{c}/dT)dT}$$
where *T*_0_ and *T*_∞_ represent the initial temperature and final temperature of the crystallization process, respectively.

Figure 3a is the curve of relative crystallinity with temperature in the crystallization process of PEG. The relative crystallinity and the crystallization temperature are obviously reverse S-shaped. Figure 3b is the curve of relative crystallinity with the temperature during the crystallization of PEG-PCL. The curve is not an obvious reverse s-shaped curve when the heating rate is 50 K/s. The reason is that the PEG block and PCL block of PEG-PCL crystallize at different temperatures. Therefore, the peak splitting behavior on the crystallization curve is obvious.

The values of *T*_0_ (initial crystallization temperature), D (crystallization temperature range), *t*_1/2_ (half-time of crystallization) and *T*_1/2_ (half-temperature of crystallization) of PEG and PEG-PCL are listed in Table 1 when the crystallization rate is 50 K/s, 100 K/s, 200 K/s, 300 K/s, 400 K/s, 500 K/s, and 600 K/s. In the entire non-isothermal crystallization process, the crystallization time and the corresponding temperature have the following relationship [12]:(2)$$t=\frac{T\;-\;T_{0}}{\Phi }$$

It is clear that as the cooling rate increases, the half crystallization time becomes shorter, which means that the crystallization rate becomes faster and faster. The half crystallization time of PEG is shorter than that of PEG-PCL, indicating that the crystallization rate of PEG is faster than that of PEG-PCL. At a cooling rate of 50–200 K/s, the crystallization temperature range of PEG-PCL is larger than those of the PEG polymer, which means that the presence of the PCL block makes the crystallization temperature range of PEG-PCL wide. The initial crystallization temperature of PEG-PCL is significantly higher than that of PEG. This is because the PCL block crystallizes first, and the crystallization temperature of PCL is higher than that of PEG. However, as the cooling rate continues to increase, the initial crystallization temperature of PEG-PCL becomes less than that of the PEG polymer. In order to explain this phenomenon, we used FDSC to study the crystallization behavior of PCL with a molecular weight of 5000, and we found that PCL no longer crystallizes when the cooling rate is 50 K/s. Therefore, we infer that when the cooling rate reaches 300 K/s, the crystallization of the PCL block is inhibited. After the cooling rate reaches 300 K/s, we find that the crystallization temperature range of PEG-PCL is approximately the same as that of PEG. This also proves from the side that only the PEG block crystallizes when the cooling rate exceeds 300 K/s. When the cooling rate is too fast, the PCL block cannot crystallize and the PCL block inhibits the crystallization of the PEG block.

In general, we found that the crystallization behavior of PEG-PCL and PEG observed by FDSC is not completely consistent with the results observed through conventional DSC. For PEG, we found that when the cooling rate reaches 1000 K/s, its crystallization behavior is inhibited. For PEG-PCL, we found that when the cooling rate is 50 K/s–200 K/s, the PCL block crystallizes first and then the PEG segment crystallizes, which is consistent with the conventional DSC observation results. However, only the PEG segment crystallizes when the cooling rate reaches 300 K/s. Neither the PEG block nor the PCL block participates in the crystallization when the cooling rate reaches 1000 K/s.

### 3.2. Non-Isothermal Crystallization Kinetic Analysis

#### 3.2.1. Ozawa Equation

According to the Ozawa model, the crystallinity at a certain temperature has the following relationship with the heating or cooling rate *Φ* [13]:(3)$$\ln\;[1-X(T)]=-\frac{k_{0}(T)}{\Phi^{m}}$$
where *k*_0_(*T*) is the crystallization rate constant, *m* is the exponent which is related to the crystal formation and nucleation mechanism where one-dimensional growth, *m* = 2; two-dimensional growth, *m* = 3; three-dimensional growth, *m* = 4, and *X*(*T*) is the relative crystallinity. It can be derived from the logarithm of both sides of the equation:(4)$$\ln\{-\ln[1-X(T)]\}=\ln k_{0}(T)-m\ln\Phi$$

According to the analysis of the Ozawa model, the graph of ln[−ln(1 − *X*(*T*))] versus ln*Φ* should show a linear relationship, with a slope exponent of *m* and an intercept of ln*k*_0_(*T*). At different temperatures *T*, the relationship between ln[−ln(1 − *X*(*T*))] and ln*Φ* analyzed by the Ozawa model is shown in Figure 4. It can be seen from Figure 4 that the Ozawa model fits the experimental data very well.

The Ozawa parameters *m*, ln*k*_0_(*T*), *k*_0_(*T*) showed in Table 2 are obtained from Figure 4. Both *m* and *k*_0_(*T*) decrease with decreasing temperature. The average value of m of PEG is 4.15, which indicates that PEG is three-dimensional spherulitic growth. The average m of PEG-PCL is 3.52, which means that the crystallization behavior and nucleation mechanism of PEG-PCL and PEG are different. However, since the crystallization behavior of PEG-PCL is related to the cooling rate; we cannot directly judge its specific crystallization behavior through the average value.

#### 3.2.2. Avrami Equation

According to the revised Avrami model, the relative crystallinity can be calculated by the following formula [12]:(5)$$X(t)=1-\exp(-Z_{t}\;t^{n})=1-\exp[-{\left(K_{\mathrm{Avrami}}\;t\right)}^{n}]$$

Transform the equation and take the logarithm on both sides to obtain
ln[−ln(1−*X*(*t*))] = *n*ln*t* + ln*Z_t_*(6)
where *Z_t_* and *n* represent the kinetic rate constant and Avrami exponent, respectively. If ln[−ln(1 − *X*(*t*))] is plotted against ln*t*, the intercept of the graph is *Z_t_*, and the slope is *n*. The Avrami equation was originally used to study the phase transition of metals, and now it is also commonly used to study the isothermal crystallization kinetics of semi-crystalline polymers. The Avrami exponent can evaluate the growth dimension and nucleation mechanism of crystalline polymers.

Figure 5 presents plots of ln[−ln(1 − *X*(*t*))] as a function of ln*t* for PEG and PEG-PCL. In Figure 5, plots of ln[−ln(1 − *X*(*t*))] versus ln*t* showed good linearity.

Table 3 summarizes the main kinetic parameters of PEG deduced from the Avrami analysis. It can be seen from the table that the value of the Avrami index n decreases with the increase in the cooling rate, which indicates that the crystallization dimension of PEG decreases with the increase in the crystallization rate. The average value of the Avrami exponent is 3.83, which indicates that the nucleation growth mode of PEG is mainly homogeneous nucleation spherulite growth. The data of non-isothermal crystallization of PEG observed by DSC also reported that its Avrami index is between 3.8 and 4.5 [16]. The crystallization rate constantly increases as the cooling rate increases. This is because the cooling rate increases and the crystallization becomes faster. However, when the temperature drop rate is 600 K/s, the decrease in the crystallization rate constant is due to the suppression of crystallization at a higher temperature drop rate.

Additionally, it shows a satisfying agreement between the reciprocal crystallization half-times (1/*t*_1/2_) that were directly determined from the experimental data and those that were deduced from the Avrami parameters. This shows that the calculated parameters can describe the non-isothermal crystallization process of PEG well.

Table 4 is a summary of the parameters of PEG-PCL. Since the crystallization behavior of PEG-PCL is different at different cooling rates, the values of n calculated according to the Avrami equation are also very different. The value of n is close to 4 when the cooling rate is 50–100 K/s, indicating that the nucleation growth mode of PEG-PCL is the homogeneous nucleation of three-dimensional spherulitic growth. When the cooling rate is 200–600 K/s, the average value of the Avrami exponent of PEG-PCL is 3, indicating that the crystallization method is mainly the growth of three-dimensional spherulites with heterogeneous nucleation. The crystallization rate constantly increases with the increase in the cooling rate, which indicates that the faster the cooling rate, the faster the crystallization.

Similarly, we compared the reciprocal crystallization half-times (1/*t*_1/2_) that were directly determined from the experimental data and those that were deduced from the Avrami parameters. It also shows good consistency, which means that the calculated parameters can describe the non-isothermal crystallization process of PEG-PCL well.

#### 3.2.3. Combined Avrami Equation and Ozawa Equation

Because there are many parameters of non-isothermal crystallization, it is difficult to describe with a single method. For this reason, Mo Zhishen et al. proposed a new method to analyze the kinetic parameters of polymer crystallization, combining the Ozawa equation and the Avrami equation to obtain the following equation [14]:(7)$$\ln Z_{t}+n\ln t=\ln k_{0}(T)-m\ln\;\Phi$$

For a certain crystallinity, the equation can be changed to:(8)lnΦ=ln F(T)−blnt
where the physical meaning of F(*T*) is the necessary cooling rate when the measured system arrives at a certain crystallinity degree at 1 min (or 1 s) crystallization time. b = n/m, n is the Avrami exponent, and m is the Ozawa exponent. Under a certain degree of crystallinity, plot ln*Φ* versus ln*t*, the slope is −b, and the intercept is lnF(*T*). The larger the F(*T*), the lower the crystallization rate of the system. Figure 6 is a plot of ln*Φ* versus ln*t*. It can be seen that the linear relationship between ln*Φ* and ln*t* is great in Figure 6.

At the same time, F(*T*) and b are summarized in Table 5. The b value of PEG and PEG-PCL increases with the increase in crystallinity. For PEG, b > 1, which means that the Avrami exponent of PEG is greater than the Ozawa exponent. For PEG-PCL, b < 1, which indicates that the Avrami exponent of PEG-PCL is smaller than the Ozawa exponent. For PEG, the F(*T*) value does not change much with the increase in crystallinity. The F(*T*) of PEG-PCL increases with the increase in crystallinity, which means that the crystallization rate of PEG-PCL decreases with the increase in crystallinity. Meanwhile, the F(*T*) of PEG is lower than that of PEG-PCL, indicating that the crystallization rate of PEG-PCL is slower than that of PEG, and this is consistent with the above results.

## 4. Conclusions

The non-isothermal crystallization behavior and isothermal crystallization behavior of PEG, and PEG-PCL were studied by FDSC. We found that both the PEG block and the PCL block of PEG-PCL can crystallize when the cooling rate is 50–100 K/s. When the cooling rate continues to increase, only the PEG block can crystallize. Neither the PEG block nor the PCL block of PEG-PCL can crystallize when the cooling rate reaches 1000 K/s. The Ozawa model and the Avarami model mainly tell us the growth dimension of crystals, while the MO model can tell us the rate of crystallization. Through the combination of the three models, we can better simulate the cooling and crystallization process in the actual machining process. The analysis of non-isothermal crystallization kinetics shows that the Ozawa, improved Avrami and Mo methods can describe the system well. The Ozawa exponent increases with increasing temperature, and the value of n is close to 4, which indicates that the crystallization process is three-dimensional growth. The Avrami exponent of PEG is 2.14 to 4.95, and its growth mode is mainly three-dimensional spherulitic growth. The Avrami exponent of PEG-PCL is close to 4 when the cooling rate is 50–100 K/s, the growth mode is mainly the three-dimensional spherulite growth with homogeneous nucleation. When the cooling rate is 200–600 K/s, the average Avrami exponent of PEG-PCL is near 3, and its growth mode is mainly three-dimensional spherulitic growth with heterogeneous nucleation. The MO model shows that the crystallization rate of PEG is greater than that of PEG-PCL.

## Figures and Tables

**Figure 1 polymers-13-03713-f001:**
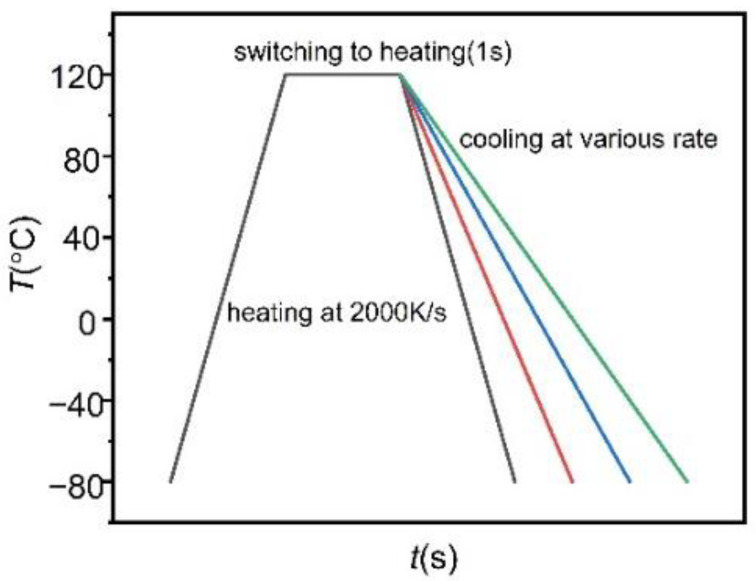
Temperature-time profiles of fast-scan measurements for the samples prepared at different cooling rates for crystallization.

**Figure 2 polymers-13-03713-f002:**
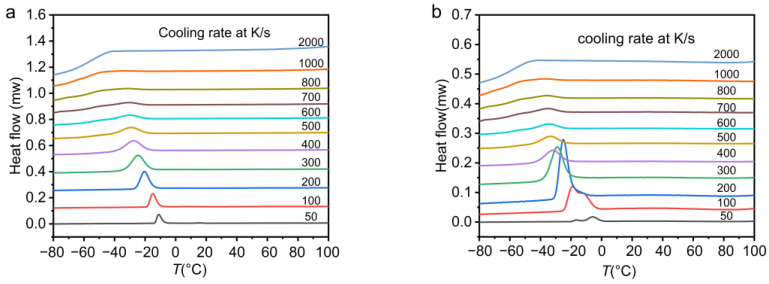
(**a**) PEG (**b**) PEG-PCL at different cooling rates of heat flow rate and temperature change cooling curve.

**Figure 3 polymers-13-03713-f003:**
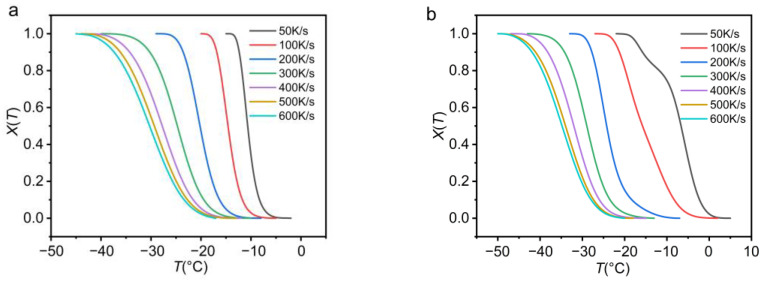
(**a**) PEG (**b**) PEG-PCL relative crystallinity curve with temperature.

**Figure 4 polymers-13-03713-f004:**
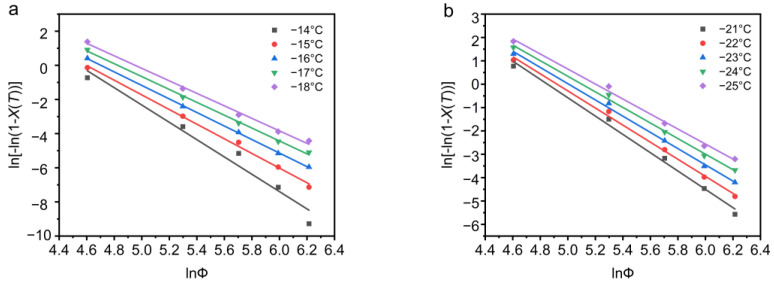
(**a**) PEG and (**b**) PEG-PCL non-isothermal crystallization process ln[−ln(1 − *X*(*T*))] plots ln*Φ*.

**Figure 5 polymers-13-03713-f005:**
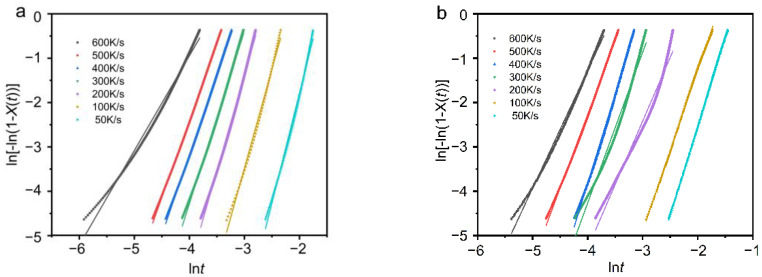
(**a**) PEG and (**b**) PEG-PCL non-isothermal crystallization ln[−ln(1−*X*(*t*))] plotting ln*t*.

**Figure 6 polymers-13-03713-f006:**
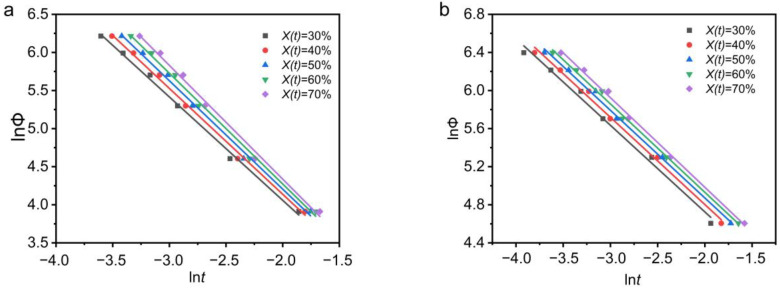
(**a**) PEG (**b**) PEG-PCL non-isothermal crystallization of ln*Φ* vs. ln*t*.

**Table 1 polymers-13-03713-t001:** *T*_0_, D(K), *t*_1/2_, *T*_1/2_ values of PEG and PEG-PCL at different cooling rates.

Cooling Rate (K/s)	PEG				PEG-PCL			
	*T*_0_ (°C)	D (°C)	*t*_1/2_ (s)	*T*_1/2_ (°C)	*T*_0_ (°C)	D (°C)	*t*_1/2_ (s)	*T*_1/2_ (°C)
50	−2	13	0.1726	−10.63	5	27	0.2340	−6.70
100	−5	15	0.0968	−14.68	2	25	0.1777	−15.77
200	−8	21	0.0605	−20.10	−7	26	0.0866	−24.32
300	−10	30	0.0487	−24.61	−13	30	0.0535	−29.06
400	−12	32	0.0393	−27.72	−15	32	0.0429	−32.17
500	−13	31	0.0323	−29.13	−18	32	0.0315	−33.74
600	−17	28	0.0221	−30.26	−20	30	0.0248	−34.87

**Table 2 polymers-13-03713-t002:** Ozawa parameters of PEG and PEG-PCL at different temperatures.

PEG			PEG-PCL		
*T*/°C	*m*	*k*_0_(*T*)	*T*/°C	*m*	*k*_0_(*T*)
−14	5.07	13.5 × 10^8^	−21	3.92	1.82 × 10^8^
−15	4.28	3.63 × 10^8^	−22	3.66	0.65 × 10^8^
−16	3.96	1.25 × 10^8^	−23	3.47	0.36 × 10^8^
−17	3.77	0.79 × 10^8^	−24	3.33	0.24 × 10^8^
−18	3.67	0.74 × 10^8^	−25	3.21	0.18 × 10^8^
$\bar{m}$ = 4.15			$\bar{m}$ = 3.52		

**Table 3 polymers-13-03713-t003:** Avrami parameters of PEG at different cooling rates.

*Φ* (K/s)	*n*	*Z_t_* (s^−n^)	k (s^−1^)	1/*t*_1/2_ (s^−1^)	(*Z_t_*/ln2)^1/n^ (s^−1^)
50	4.95	3.36 × 10^3^	5.16	5.79	5.55
100	4.45	1.99 × 10^4^	9.25	10.33	10.04
200	4.32	1.13 × 10^5^	14.78	16.53	16.09
300	3.89	7.92 × 10^4^	32.84	20.53	19.96
400	3.61	7.92 × 10^4^	43.05	25.44	25.18
500	3.46	9.20 × 10^4^	52.92	30.96	30.24
600	2.14	2.00 × 10^3^	34.87	45.25	41.39

**Table 4 polymers-13-03713-t004:** Avrami parameters of PEG-PCL at different cooling rates.

*Φ* (K/s)	*n*	*Z_t_* (s^−n^)	k (s^−1^)	1/*t*_1/2_ (s^−1^)	(*Z_t_*/ln2)^1/n^ (s^−1^)
50	4.03	2.47 × 10^2^	3.92	4.27	4.29
100	3.62	3.87 × 10^2^	5.19	5.63	5.73
200	2.87	4.83 × 10^2^	8.61	11.55	9.78
300	3.40	1.13 × 10^4^	15.56	18.69	17.48
400	4.02	2.11 × 10^5^	21.11	23.31	23.12
500	3.31	5.93 × 10^4^	27.67	31.75	30.91
600	2.65	1.13 × 10^4^	33.84	40.32	38.86

**Table 5 polymers-13-03713-t005:** Values of PEG and PEG-PCL parameters under Mo model.

*X*(*T*)	PEG			PEG-PCL		
	b	F(*T*)	r^2^	b	F(*T*)	r^2^
30%	1.36	3.8574	0.99642	0.91	18.3568	0.99171
40%	1.39	3.8962	0.99625	0.92	19.4919	0.99519
50%	1.42	3.8962	0.99566	0.92	20.6972	0.99718
60%	1.45	3.8962	0.99494	0.93	21.5419	0.99746
70%	1.49	3.8574	0.99409	0.94	22.1980	0.99704

## Data Availability

Not applicable.

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
