# Peer review of "Non-Isothermal Crystallization Kinetics of Poly(ethylene glycol) and Poly(ethylene glycol)-B-Poly(ε-caprolactone) by Flash DSC Analysis"

_polymers, 2021, doi:10.3390/polym13213713_

Round 1
Reviewer 1 Report
This is an interesting study, however, for people not dealing with this specialised DSC technique, the introduction is missing important information. How does the technique work, what are its advantages? For regular DSC experiments, the heating usually proceeds at 10deg/s. You are using 2000K/s? How does this work? Can you describe the technique a little better in the introduction?
What is the size/weight of the sample measured by DSC?
p.1 line 32 Omit this part of the sentence, makes no sense: "which has a great effect on the study of crystallization kinetics".
The text in 2.2 is not about sample preparation, change the title, sample preparation text should be added too though.
In the conclusions, explain a little better why you combined the three models. What they tell you separately and together. How can you use the obtained knowledge?
Reviewer 2 Report
- The Introduction of the paper is very general. The authors only outline the characteristics of their method of measurement using the FDSC technique and how this compares with the conventional DSC. They also only mention about the advantages of using PEG and PEG-PCL in applications and why their knowing of crystallization properties is important. The authors perhaps should mention also about a few previous works from the literature in which specifically the crystallization behavior of PEG and PEG-PCL was studied.
- The time units are missing from the horizontal (time) axis of the graph in Figure 1.
- The authors use for the temperature units throughout the paper both Kelvin and Celcius units. I am wondering if for the sake of uniformity in the units of the mentioned physical quantities they should only use one of the two units throughout the paper (?).
- It is not clear whether the results that the authors present in Figures 2 and 3 and list on Table 1 correspond to measurements using the conventional DSC technique or the FDSC? - the authors mention in lines 127-129 that in order to explain their observations they used FDSC thus it is not clear whether the presented results are from DSC or FDSC measurements (?).
